# High-protein diets reduce plasma pro-inflammatory cytokines following lipopolysaccharide challenge in Swiss Albino mice

Hellen W. Kinyi[1,2]*, Charles Kato Drago[3], Lucy Ochola[4], Gertrude N. Kiwanuka[1]

1 Department of Biochemistry, Faculty of Medicine, Mbarara University of Science and Technology, Mbarara, Uganda, 2 Department of Biomedical Sciences, Medical College East Africa, The Aga Khan University, Nairobi, Kenya, 3 School of Biosecurity, Biotechnical and Laboratory Studies, College of Veterinary Medicine, Animal Resource and Biosecurity, Makerere University Kampala, Uganda, 4 Department of Tropical and Infectious Diseases, Institute of Primate Research (IPR), Karen, Kenya

* hellen.kinyi@aku.edu

## Abstract

Macronutrients serve as principal sources of energy, structural components, and regulators of physiological processes. However, the optimal macronutrient combination for health remains unclear. While previous studies indicate that dietary macronutrient composition influences immune function, many have examined individual nutrients in isolation, failing to reflect the interactive effects of macronutrients. This study addresses this gap by examining how varying ratios of dietary carbohydrates, proteins, and lipids modulate serum cytokine responses to lipopolysaccharide challenge in Swiss albino mice. Male and female Swiss albino mice (n = 6 per group), aged 6–8 weeks, were randomly assigned to six purified isocaloric diets with differing macronutrient ratios for 15 weeks. Body weights were monitored to assess nutritional status. Serum levels of TNF-α, IL-6, IL-1β, and IL-10 were measured in unchallenged mice and after three hours of intraperitoneal LPS administration. Mice fed high-carbohydrate, low-protein diets had the highest weight (33.1 g ± 1.1), while those on high-lipid, low-protein diets had the lowest (28.3 g ± 0.6). Plasma levels of TNF-α and IL-10 varied significantly (p < 0.05) by diet in the unchallenged mice. IL-1β did not differ markedly (p = 0.085) across the dietary groups, and IL-6 levels were below the assay's detection limit (<230.312 pg/mL). Following the lipopolysaccharide challenge, all cytokines increased, with significant differences among diets. Mice on high-protein diets exhibited notably lower TNF-α, IL-6, and IL-1β levels compared to those on high-carbohydrate or high-lipid diets. In contrast, IL-10 levels were higher in mice fed low-protein, high-carbohydrate, or high-lipid diets. In conclusion, high-protein diets appeared to dampen the responsiveness to lipopolysaccharide challenge, as indicated by smaller increases in pro-inflammatory cytokine levels, whereas high-carbohydrate and high-lipid diets elicited greater cytokine responses. We recommend

**Data availability statement:** All relevant data are within the manuscript and its Supporting Information files.

**Funding:** The author(s) received no specific funding for this work.

**Competing interests:** The authors have declared that no competing interests exist.

that nutritional strategies aimed at modulating inflammation should ensure adequate dietary protein to help protect against both acute and chronic inflammation.

## 1 Introduction

Macronutrients, including proteins, fats, and carbohydrates, are principal components of diets and hence consumed in large quantities [1]. Functionally, they serve as principal sources of energy and structural components of the body and are important for physiological processes such as growth, development, and immune function [2,3]. Although these macronutrients are required for health, the combination that promotes optimal health has remained elusive [4]. This could be due to the complex interactions between nutrients and other food constituents within an organism's physiology, as well as differing nutritional requirements for physiological traits such as reproduction, growth, and immunity [5].

The immune system is an elaborate network of cells and molecules that protects the host from pathogens [6]. Previous research shows that dietary macronutrient composition significantly influences immune function in mammals. Different macronutrient ratios optimize distinct immune traits, suggesting trade-offs between immune defense and other physiological demands like growth and reproduction [7]. In addition, macronutrients modulate both innate and adaptive immunity by affecting immune cell signaling, proliferation, differentiation, and antibody production [8]. They also serve as fuels and modulators of immune signaling pathways [9]. For example, neutrophils have increased demands for glucose to sustain glycolysis, while activated macrophages rely on cholesterol derivatives for membrane remodeling [10].

The innate immune arm recognizes conserved pathogen-associated molecular patterns (PAMPs), such as lipopolysaccharide (LPS), through pattern recognition receptors like Toll-like receptor 4 (TLR4) [11]. Their interaction activates signaling cascades culminating in activation of nuclear factor kappa B (NF-κB) and subsequent production of cytokines such as tumor necrosis factor alpha (TNF-α), interleukin-6 (IL-6), interleukin-10 (IL-10), and interleukin-1β (IL-1β) which cause either inflammatory or anti-inflammatory responses [12].

Emerging evidence shows that nutritional status and dietary composition can regulate these immune responses. Imbalances in macronutrient intake have been linked to altered immune responses, for example, obesity is associated with increased levels of pro-inflammatory cytokines such as TNF-α, IL-6, and IL-1β, while undernutrition tends to elevate anti-inflammatory cytokines like IL-10 [13,14]. Furthermore, dietary macronutrient content influences cytokine production in both human and murine studies. For example, high-fat and carbohydrate-rich diets are reported to promote pro-inflammatory cytokine expression, while low-protein diets may impair the production of both pro- and anti-inflammatory cytokines [15,16]. Moreover, micronutrients such as zinc and vitamin D have been implicated in regulating cytokine production, indicating the complex interplay between diet and immune function [17]. However, there is scarce information on how distinct macronutrient ratios modify cytokine responses to LPS challenge.

While caloric restriction and manipulation of individual macronutrients have been studied for their effects on health and longevity, the interactive effects of macronutrient combinations remain poorly understood [18]. Many of these studies alter one nutrient at a time, which fails to reflect the complexity of real-world diets. Evidence suggests that varying macronutrient ratios could influence immune function and inflammation, suggesting a critical need for studies assessing their combinatorial effects [19]. Although some studies have used extensive diet grids to explore these interactions, few have evaluated how defined variations in macronutrient ratios influence innate immune responses. This study therefore aimed to determine whether macronutrient ratios predict cytokine response following LPS challenge in Swiss albino mice.

## 2 Materials and methods

### 2.1 Diets

Six isocaloric diets (3.8 kcal/g) were formulated using food-grade ingredients, based on the American Institute of Nutrition-1993 maintenance diet (AIN-93M) guidelines [20,21]. The standard AIN-93M diet provides approximately 76% carbohydrate, 15% protein, and 9% lipid by weight [22]. The local formulation used has been validated in our previous work with Swiss albino mice [23]. The source of protein was casein supplemented with cysteine, soybean oil for lipids, and a carbohydrate mix of corn-starch, sucrose, and maltodextrin with added cellulose, vitamin, and mineral mixes according to AIN-93M guidelines. Based on lifespan-optimizing models, experimental diets initially adopted macronutrient ratios of 5% protein, 75% carbohydrate, and 20% lipids [24]. However, severe weight loss (>40%) in both sexes and mortality in the high lipid low protein (HLLP) group prompted dietary optimization: protein was increased to 8% in low-protein diets and reduced to 60% in high-protein diets. Final macronutrient ratios are provided in Table 1.

### 2.2 Experimental design

**2.2.1 Mice.** Swiss Albino mice were selected for this study due to their high genetic similarity to humans, with about 95–98% of their genes having human counterparts, and their comparable physiological responses [25] Click or tap here to enter text. They are also easy to maintain, have a small body size, are adaptable to various diets, and have well-characterized cytokine and TLR4 responses that are relevant to LPS-induced immune activation [26,27].

**2.2.2 Sample size.** The sample size was initially estimated using LaMorte's Power Calculator (Boston University, Microsoft Excel spreadsheet) based on the power analysis method. Input variables for expected effect size and standard deviation were obtained from data on body weight changes in mice fed an AIN-93 diet versus a high-fat diet for eight weeks [28]. Using a power of 80% and a significance level p-value of 0.05, the estimated number was 10 mice per group. However, in accordance with the ethical principles of the 3Rs (Replacement, Reduction, and Refinement) to minimize animal use while still achieving statistically meaningful outcomes, a final sample size of 6 mice per group (3 males and 3 females) was adopted. This provided sufficient replication for comparative statistical analyses while reducing animal use.

**Table 1. Percentage composition of macronutrients in experimental diets.**

| Experimental group | HCLL | HCLP | HPLC | HPLL | HLLC | HLLP |
|---|---|---|---|---|---|---|
| CHO | 75 | 72 | 10 | 30 | 5 | 20 |
| Protein | 20 | 8 | 60 | 60 | 20 | 8 |
| Lipid | 5 | 20 | 30 | 10 | 75 | 72 |
| Total | 100 | 100 | 100 | 100 | 100 | 100 |

Macronutrient values are expressed as percentage of total dietary energy (% kcal) contributed by carbohydrate, protein, and lipid. All diets were isocaloric at 3.8 kcal/g. Diets composition adapted from American Institute of Nutrition 1993 maintenance diet (AIN-93M) [23]. CHO: carbohydrates. Group codes of macronutrient ratios in the diets: **HCLL** – High Carbohydrate Low Lipid; **HCLP** – High Carbohydrate Low Protein; **HPLC** – High Protein Low Carbohydrate; **HPLL** – High Protein Low Lipid; **HLLC** – High Lipid Low Carbohydrate; **HLLP** – High Lipid Low Protein.

**2.2.3 Experimental design.** Swiss albino mice (male and female) aged 6–8 weeks and weighing 18–23 g (average weight 19.9±0.2 g SEM) were used in this study. Only healthy animals within this age and weight range were included. Eligible mice were then randomly assigned to six dietary groups, each comprising 12 animals (6 males and 6 females), and fed diets with distinct macronutrient ratios (Table 1). Randomization was achieved by sequentially allocating animals that fulfilled the inclusion criteria to each group in turn (i.e., the first animal to group 1, the second to group 2, and so on until all six groups were filled, then repeating the cycle until each group contained 6 males and 6 females). This ensured balanced distribution of sex and weight across groups while minimizing selection bias. The animals were maintained on experimental diets for 15 weeks, as sustained feeding beyond 12 weeks is required to observe impacts of diets [29]. A schematic representation of the experimental timeline is provided in Fig 1.

**2.2.4 Husbandry and humane endpoint.** Mice were housed in metabolic cages (35×30×15 cm) under ambient conditions (20–25 °C, 70–80% humidity), with a 12-hour light/dark cycle. Fresh water and food were provided ad libitum daily, while cage cleaning and replacement of bedding were done weekly to maintain hygiene. Male and female mice were housed separately to prevent breeding. Each group of mice received 5 g of food pellets per mouse per day, as previously described [23]. Food consumption was monitored twice daily, and food was replaced each morning with a fresh batch after inspection. Only minimal leftover pellets were occasionally observed, suggesting that the mice consumed nearly all the food provided in each group. To minimize stress and variability in measurements, all animals were handled consistently, either by gentle cupping in the palm or by brief restraint at the base of the tail when necessary. Animals were observed twice daily throughout the experiment for general health, activity, and behavior.

Humane endpoint criteria were defined prior to the start of the study in order to minimize animal suffering. These included severe lethargy, inability to access food or water, pronounced piloerection persisting beyond 12 hours, rapid or labored breathing, unresponsiveness to gentle stimulation, or a reduction in body weight exceeding 20% of baseline. If any of these criteria were met, the affected animal would have been humanely euthanized within 24 hours by intraperitoneal injection 100µl of Ketamine and Xylazine (respectively 80 mg/kg and 10 mg/kg), in Phosphate Buffer Saline (PBS). However, no animals died unexpectedly or exhibited signs of severe distress during the 15 weeks of feeding, and none reached humane endpoint criteria.

**2.2.5 Ethics approval.** All procedures followed the guidelines for the care and use of laboratory animals [30] and were approved by the Mbarara University Institutional Ethics Committee (Study No. 19/08–20). The study registered by the Uganda National Council for Science and Technology (NS159ES). The researchers staff underwent a 13-week structured training in laboratory animal science prior to commencement of experimental procedures. The training was facilitated by an expert in the use of animals in research at the Department of Pharmacology, Mbarara University of Science and Technology. The sessions covered ethical aspects of animal experimentation and relevant legislation, biology and husbandry of laboratory animals, animal behavior and welfare, standardization of animal experimentation, nutrition, genetic and microbiological standardization, animal diseases, concepts of reduction and replacement in animal models, phases of animal experimentation, design and execution of experiments, organizational and management aspects, and procedures relating to anesthesia, analgesia, euthanasia, and recognition of pain and distress.

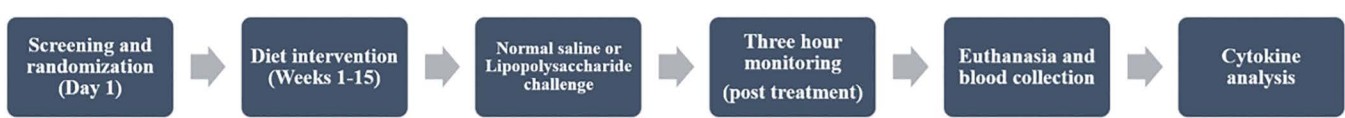

**Fig 1. Experimental timeline.** Swiss albino mice were screened for inclusion criteria and randomized into six dietary groups (6 males and 6 females per group). Mice were maintained on the respective diets for 15 weeks. At week 15, each dietary group was subdivided into two: a control group receiving normal saline and a treatment group receiving lipopolysaccharide. Animals were monitored for 3 hours post-treatment, after which they were euthanized and blood samples collected for cytokine analysis.

## 2.3 Lipopolysaccharide challenge

After 15 weeks of feeding six mice (3 males and 3 females) were randomly selected to receive an intraperitoneal injection of 5 mg/kg lipopolysaccharide (LPS) from *Escherichia coli O111:B4* (Sigma-Aldrich) [31,32]. LPS was used to induce acute systemic inflammation by activating Toll-like receptor 4 (TLR4) on immune cells, leading to rapid cytokine production [33]. The remaining six mice (3 males and 3 females) received an intraperitoneal injection of 1 ml of normal saline and served as negative controls.

At three hours post-treatment, terminal anesthesia was induced by intraperitoneal administration of xylazine (16 mg/kg) and ketamine (60 mg/kg) [34]. Depth of anesthesia was confirmed by the absence of the pedal withdrawal reflex before any procedure was initiated. While under deep anesthesia, approximately 1 ml of blood was collected by cardiac puncture. Animals were euthanized by exsanguination under anesthesia, and death was confirmed by cessation of heartbeat and respiration. Whole blood was collected into ethylenediamine tetraacetic acid (EDTA)-coated tubes, centrifuged at 1,500 × g for 15 minutes, and plasma stored at −80 °C until analysis. Animal carcasses were collected and disposed of by incineration together with other biological waste materials in accordance with institutional biosafety procedures.

## 2.4 Cytokine analysis

Plasma concentrations of IL-6, IL-10, IL-1β, and TNF-α were quantified using a premixed multiplex Luminex® Discovery Assay (Mouse Premixed Multi-Analyte Kit; R&D Systems, USA) on the Luminex® 100/200™ platform (Luminex Corporation, TX, USA), following the manufacturer's instructions. All assays were performed in duplicate.

Briefly, after all reagents were thawed to room temperature and prepared according to the kit insert, 50 μl of standards and plasma samples were added to their respective wells in duplicate on a pre-planned plate layout. Next, 50 μl of diluted micro particle cocktail was added to each well. The 96-well microplate was covered with a foil plate sealer and incubated for 2 hours at room temperature on a horizontal orbital microplate shaker at 800 rpm.

Following incubation, wells were washed three times by adding 100 μl of wash buffer, allowing it to stand for 1 minute, and then discarding it. Subsequently, 50 μl of diluted biotin antibody cocktail was added to each well, and the plate was incubated for 1 hour at room temperature on a shaker at 800 rpm. After another three washes, 50 μl of diluted streptavidin–phycoerythrin (PE) was added to each well and incubated for 30 minutes at room temperature with shaking. A final wash step was performed, and 100 μl of wash buffer was added to each well, followed by a 2-minute incubation on the shaker.

The plate was then inserted into the Luminex® 100/200™ instrument, and fluorescence detection of bead-bound cytokine was acquired within 90 minutes. Calibration curves were generated using standard analyte concentrations run concurrently to ensure assay accuracy and precision.

## 2.5 Data analysis

Results from each experiment were recorded in an Excel spreadsheet and subsequently imported into the Paleontological Statistical (PAST) version 4:03 for subsequent analysis. The serum cytokine levels for each group were tested for normality using the Shapiro-Wilk test to determine whether parametric or non-parametric statistical methods would be appropriate. Given that the data were normally distributed, they were expressed as mean ± standard error of the mean (SEM) and analysed using parametric tests. Independent samples t-tests were conducted to compare mean cytokine concentrations between male and female mice within each diet group. There were no significant sex differences, hence the data was pooled together for subsequent analysis.

Body weight results were analyzed using one-way ANOVA. First, it was used to compare initial body weights (Day 1) across the six diet groups, confirming that all groups started at comparable weights. Second, at the end of the experiment (Week 15), a separate one-way ANOVA was performed to examine differences in final body weights among the diet

groups. Where significant differences were found, pairwise comparisons were conducted using Tukey's pairwise post-hoc test to identify specific group differences. Additionally, paired t-tests were performed within each diet group to compare body weights at Day 1 versus Week 15, assessing the effect of time on weight gain within each diet.

For the cytokine analyses, one-way ANOVA tests were conducted separately for each cytokine to compare mean plasma levels across the six dietary groups. This was done both for samples collected prior to (baseline levels) and after the LPS challenge. Where significant differences were found, Tukey's multiple comparison tests were used to identify which groups differed.

Additionally, linear regression analysis was used to evaluate whether dietary macronutrient composition predicted cytokine responses. Change in serum cytokine levels (Δ cytokine = Post LPS challenge level − Pre LPS challenge level) was calculated for each mouse. Macronutrient intake was expressed as a percentage of total calories (% carbohydrate, % protein, % lipid). To prevent multicollinearity % protein and % lipid were selected as independent variables in the regression model. A p-value < 0.05 was considered statistically significant.

## 3 Results

### 3.1 Weight of the animals

There was no significant difference in the mean body weights across the six diet groups on Day 1 (ANOVA $F_{(5, 66)}$ = 1.834, p = 0.12). After 15 weeks of feeding the animals on experimental diets, there was an increase in their average weight (31.2g ± 0.5). One-way ANOVA revealed statistically significant differences in mean body weight among the groups ($F_{(5, 66)}$ = 5.491, P = 0.0003).

Mice fed on a high-carbohydrate, low-protein (HCLP) diet had the highest average weight (33.1 g ± 1.1), while the lowest average body weight was observed in mice fed on a high-lipid, low-protein (HLLP) diet (28.3g ± 0.6) as shown in Fig 2. Tukey's post-hoc analysis revealed that mice on the HLLP diet had significantly lower body weights compared to those

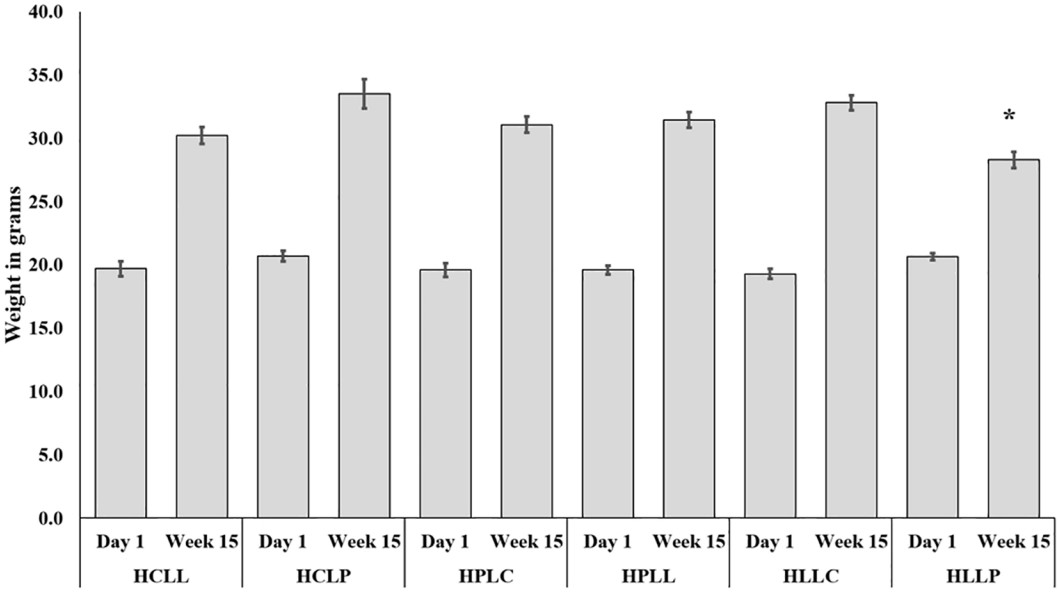

**Fig 2. Body weights of mice fed the experimental diets for 15 weeks.** Bar graphs show the mean ± SEM body weight of mice fed on experimental diets on day 1 and week 15 (n = 12 per group; 6 males and 6 females). * Significantly lower than weight of mice fed on all other experimental diets for 15 weeks. **Key: HCLL**-High Carbohydrate Low Lipid, **HCLP**-High Carbohydrate Low Protein, **HPLC**-High Protein Low Carbohydrate, **HPLL**-High Protein Low Lipid, **HLLC**-High Lipid Low Carbohydrate, **HLLP**-High Carbohydrate Low Protein.

on the HCLP (p = 0.0004), HPLC (p = 0.045), and HPLL (p = 0.050) diets. Within each diet group, paired t-tests showed a significant increase in body weight from Day 1 to Week 15 (p < 0.05 for all groups).

### 3.1 Effect of experimental diets on plasma cytokine levels

Cytokine concentrations (IL-6, TNF-α, IL-10, and IL-1β) were measured in plasma samples from mice in the six dietary groups. Data were assessed for normality using the Shapiro–Wilk test and were found to be normally distributed. One-way ANOVA, followed by Tukey's post hoc test, was used to assess differences in cytokine levels among the groups.

ANOVA showed significant differences in mean serum TNF-α levels among unchallenged mice fed the experimental diets for 15 weeks $F_{(5, 30)} = 9.037$, $p = 2.5 \times 10^{-5}$. Post hoc analysis showed that TNF-α levels were significantly higher in mice fed the HCLP diet compared to HPLC (p = 0.0006) and HPLL (p = 0.003), and in HLLP-fed mice compared to HPLC (p = 0.0004) and HPLL (p = 0.002), as shown in Fig 3. In unchallenged mice, IL-6 levels were below the assay's detection limit (<230.312 pg/mL) across all dietary groups. In contrast, IL-1β was present in the serum of all mice fed the experimental diets; however, there was no significant difference in IL-1β levels across the six groups ($F_{(5, 30)} = 2.163$, p = 0.085). ANOVA showed a significant difference in baseline serum IL-10 levels across the dietary groups ($F_{(5, 30)} = 5.599$, p = 0.0009). Mice fed the HLLP diet had the highest IL-10 levels (31.2 pg/mL ± 2.6), while those fed the HPLC diet had the lowest (16.7 pg/mL ± 1.2). Post hoc analysis showed that IL-10 levels were significantly higher in mice fed HCLL compared to HPLL (p = 0.0009). In addition, IL-10 levels were significantly higher in mice fed HCLP (p = 0.026), HLLC (p = 0.003), and HLLP (p = 0.0009) diets compared to HPLC.

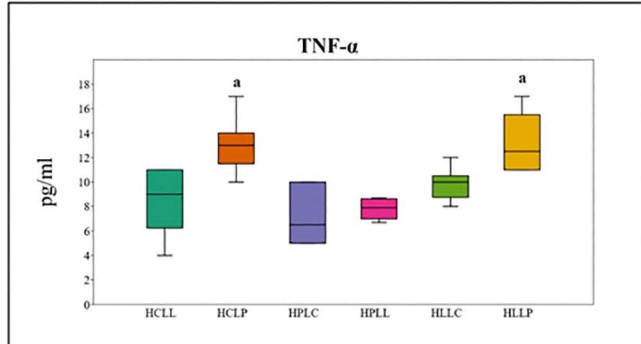
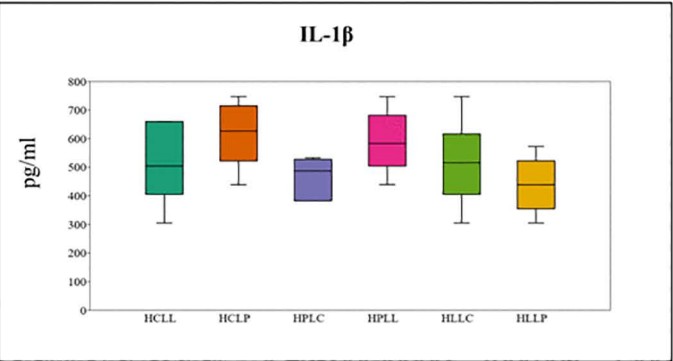
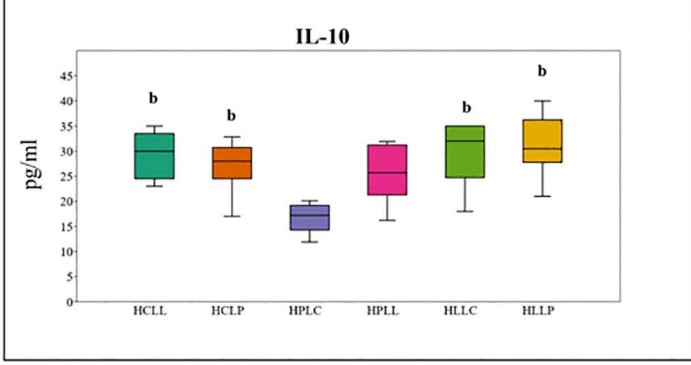

**Fig 3. Plasma cytokine levels in unchallenged mice fed experimental diets.** Box-and-whisker plots show mean ± SEM serum concentrations (pg/ml) of TNF-α, IL-1β, and IL-10, measured 3 hours after intraperitoneal injection of 1 ml of Normal saline (n = 6, 3 males and 3 females). [a] had significantly higher mean TNF-alpha levels than HPLC and HPLL. [b] had significantly higher mean IL-10 levels than HPLC. **Key: HCLL**-High Carbohydrate Low Lipid, **HCLP**-High Carbohydrate Low Protein, **HPLC**-High Protein Low Carbohydrate, **HPLL**-High Protein Low Lipid, **HLLC**-High Lipid Low Carbohydrate, **HLLP**-High Carbohydrate Low Protein.

## 3.2 Effect of LPS challenge on plasma cytokine levels in mice fed experimental diets

There was a significant increase in plasma cytokine levels measured 3 hours after LPS administration ($p < 0.05$; paired t-tests) as shown in Fig 4. Plasma TNF-α levels differed significantly across the six dietary groups ($F_{(5, 30)} = 43.66$, $p = 7.02 \times 10^{-13}$). Mice fed the HLLC diet had the highest TNF-α levels (280.7 pg/mL ± 21.9). Mice fed high-protein diets (HPLC and HPLL) had significantly lower TNF-α levels compared to those on low-protein, high-carbohydrate, or high-lipid diets. Specifically, HPLC TNF-α levels were significantly lower than those in HCLL ($p = 0.003$), HCLP ($p = 7.02 \times 10^{-15}$), HLLC ($p = 1.06 \times 10^{-11}$), and HLLP ($p = 0.0009$). Similarly, HPLL was significantly lower than HCLL ($p = 8.06 \times 10^{-5}$), HCLP ($p = 1.8 \times 10^{-6}$), HLLC ($p = 7.6 \times 10^{-13}$), and HLLP ($p = 2.3 \times 10^{-5}$), as shown in Fig 4.

IL-6 levels also differed significantly across the six dietary groups ($F_{(5, 30)} = 93.44$, $p = 2.31 \times 10^{-17}$). Mice fed the HCLL diet had the highest IL-6 levels (6602.8 pg/mL ± 182.2), while those on the HPLL diet had the lowest (2228.2 pg/mL ± 104.6). HPLL mice had significantly lower IL-6 levels compared to HCLL ($p = 3.9 \times 10^{-14}$), HCLP ($p = 3.5 \times 10^{-6}$), HPLC ($p = 0.005$), HLLC ($p = 3.1 \times 10^{-6}$), and HLLP ($p = 5.7 \times 10^{-14}$). Additionally, HPLC was significantly lower than HCLL ($p = 2.3 \times 10^{-13}$) and HLLP ($p = 4.3 \times 10^{-11}$).

For, IL-1β plasma levels increased significantly in all mice three hours after LPS challenge ($F_{(5, 30)} = 17.41$, $p = 4.43 \times 10^{-8}$). The highest mean IL-1β concentration was observed in those fed the HCLP diet (2585.3 pg/mL ± 157), while the lowest was in those fed HPLL (1106.0 pg/mL ± 127). Mice in HPLC had significantly lower IL-1β levels compared

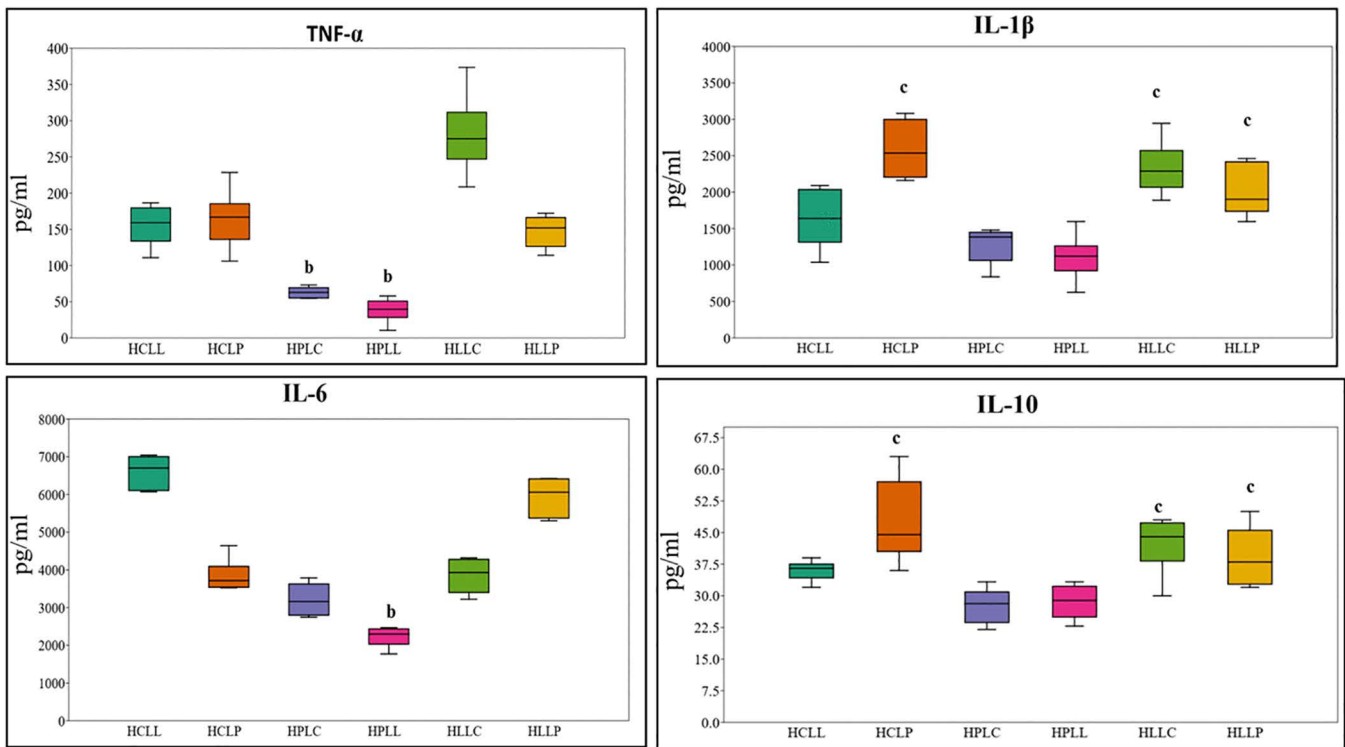

**Fig 4. Plasma cytokine levels in mice fed experimental diets following lipopolysaccharide challenge.** Box-and-whisker plots show mean ± SEM serum concentrations (pg/ml) of TNF-α, IL-1β, IL-6 and IL-10, measured 3 hours after intraperitoneal injection of 5 mg/kg of LPS (n = 6, 3 males and 3 females). [b] had significantly lower mean cytokine levels compared to those fed other diets. [c] had significantly higher mean cytokine levels than HPLC AND HPLL. **Key: HCLL**-High Carbohydrate Low Lipid, **HCLP**-High Carbohydrate Low Protein, **HPLC**-High Protein Low Carbohydrate, **HPLL**-High Protein Low Lipid, **HLLC**-High Lipid Low Carbohydrate, **HLLP**-High Carbohydrate Low Protein.

to HCLP ($p = 3.9 \times 10^{-6}$), HLLC ($p = 0.0001$), and HLLP ($p = 0.012$). Likewise, those in HPLL had significantly reduced IL-1β compared to HCLP ($p = 3.9 \times 10^{-7}$), HLLC ($p = 1.2 \times 10^{-5}$), and HLLP ($p = 0.001$) as shown in Fig 4.

Similarly, IL-10 concentrations varied significantly across the six dietary groups ($F_{(5, 30)} = 9.427$, $p = 1.76 \times 10^{-5}$). The highest IL-10 levels were observed in mice fed the HLLC diet (42.3 pg/mL ± 3.0), while the lowest were recorded in the HPLC group (27.7 pg/mL ± 2.0). HPLC mice had significantly lower IL-10 levels compared to HCLP ($p = 6.9 \times 10^{-5}$), HLLC ($p = 0.003$), and HLLP ($p = 0.034$). HPLL mice had also significantly lower IL-10 levels than those fed on HCLP ($p = 0.0001$) and HLLC ($p = 0.007$).

### 3.3 Changes in cytokine concentrations following LPS challenge

Marked differences in cytokine responses were observed among the dietary groups 3 h after LPS administration (Fig 5). Diets with high protein content (HPLC and HPLL) exhibited lower responses for TNF-α (55.7 ± 3.0 pg/ml and 30.6 ± 4.0 pg/ml, respectively) and IL-6 (3205.8 ± 169.7 pg/ml and 2228.2 ± 104.6 pg/ml, respectively). The HLLC diet had the greatest increase in TNF-α (270.8 ± 21.7 pg/ml) and IL-1β (1849.0 ± 131.5 pg/ml) while HCLL showed the largest change in IL-6 (6602.8 ± 182.2 pg/ml). IL-10 changes remained modest across all diets.

### 3.3 Interactive effects of diets and serum cytokine levels

To assess the effect of dietary macronutrient composition on cytokine responses, multiple linear regressions were performed using dietary protein and lipid content as predictors. Carbohydrates were excluded from the models to avoid

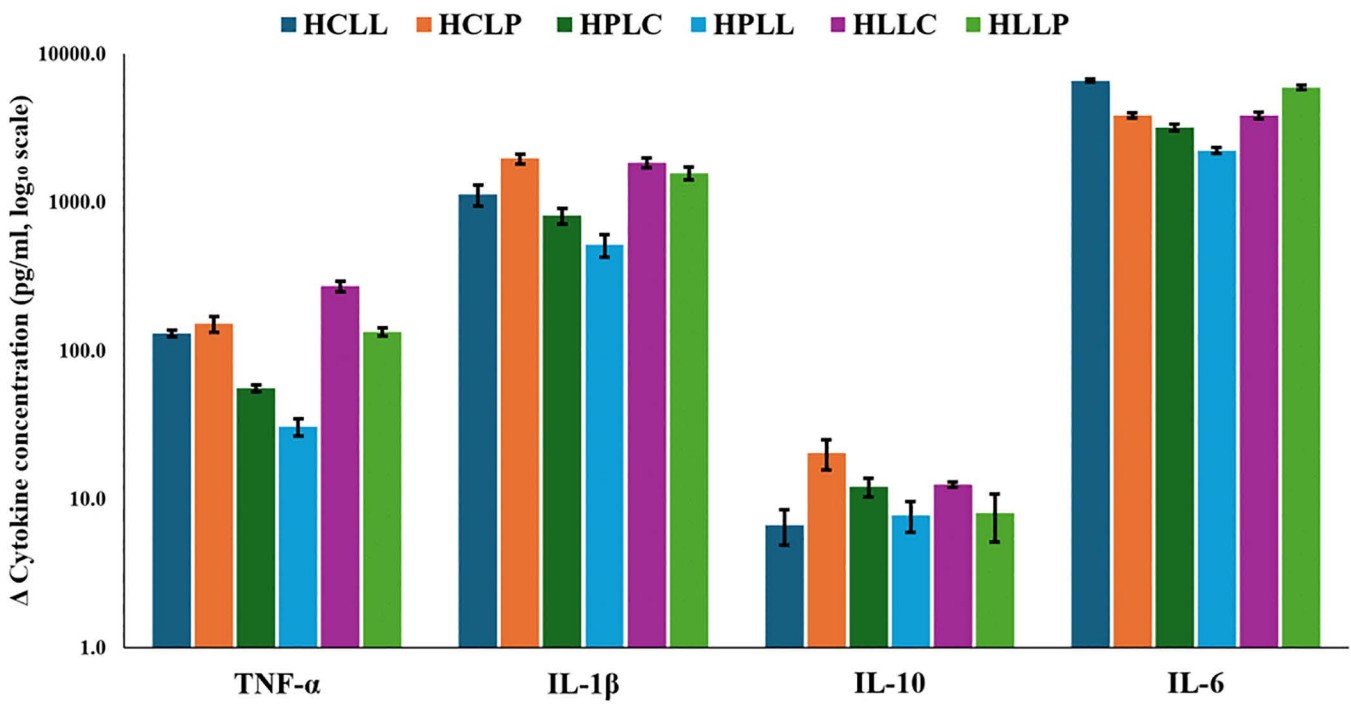

**Fig 5. Mean change in plasma cytokine concentrations following LPS challenge.** Mean (± SEM) change in TNF-α, IL-1β, IL-6, and IL-10 levels 3 h after LPS administration in mice fed different diets. Changes were calculated as post- minus pre-LPS concentrations. Data are shown on a $\log_{10}$ scale. **Key: HCLL**-High Carbohydrate Low Lipid, **HCLP**-High Carbohydrate Low Protein, **HPLC**-High Protein Low Carbohydrate, **HPLL**-High Protein Low Lipid, **HLLC**-High Lipid Low Carbohydrate, **HLLP**-High Carbohydrate Low Protein.

multicollinearity, as macronutrient percentages are compositional and sum to 100%. Including all three macronutrients would introduce perfect linear dependence, violating the assumptions of multiple regression.

The multiple regression to predict the fold-change in TNF-α, IL-6, and IL-1β from dietary protein and lipid content was significant (Table 2). The regression analysis showed that increased protein intake significantly reduced fold-changes in TNF-α, IL-6, and IL-1β. Conversely, increasing lipid intake elevated TNF-α but had no significant effect on IL-6 or IL-1β. IL-10 levels were unaffected by either macronutrient.

## 4  Discussion

In recent years, researchers have increasingly recognized the interplay between diet and immune system regulation [35–38]. While previous studies have explored the individual effects of specific macronutrients on immune response, our study takes a comprehensive approach by examining the interactive effects of carbohydrates, lipids, and proteins on cytokine production. The experimental diets parallel with certain human dietary patterns. For example, the HLLC diet resembles a ketogenic diet, which is used for weight management and treatment of epilepsy [39]. The HCLP diet is similar to cereal-based diets common in some low-income areas, where protein intake is often inadequate [40]. Conversely, high-protein diets such as HPLC reflect regimens adopted for weight loss or athletic performance [41]. By aligning these dietary models with known human consumption patterns, our study provides translational insights into how macronutrient distributions may shape inflammation, which is central to the pathogenesis and progression of chronic diseases such as cancer.

To assess effects on cytokine production, Swiss albino mice were fed isocaloric diets with differing macronutrient ratios for 15 weeks, after which they were challenged with LPS. Plasma cytokine levels were measured three hours post-injection. All animals gained weight over the study period, showing the diets provided adequate nutrition. Following LPS administration, cytokine levels increased across all dietary groups. Mice fed on high-protein diets had consistently lower plasma cytokine levels than those fed on low-protein diets paired with either high carbohydrate or high lipid, indicating a dampened cytokine response.

Mice fed on all six isocaloric diets gained weight over the 15-week period, indicating that the animals were not deficient in energy. Although the HLLP group showed a slightly lower final weight compared to other groups, the difference was modest and remained within the expected physiological range for Swiss Albino mice [42,43]. These findings suggest that the immune changes observed are likely due to dietary macronutrient composition rather than caloric variation.

### Cytokine responses before and after LPS challenge

TNF-α, IL-1β, and IL-10 were detected in plasma prior to LPS stimulation, consistent with previous reports in both humans and rodents [44–47]. These cytokines have been shown to play roles in immune surveillance, tissue maintenance,

Table 2.  Multiple linear regression analysis of dietary protein and lipid effects on LPS-induced cytokine responses.

| Cytokine | Predictor | β (95% CI) | p-value | Model Fit (R²) | F-statistic | Model p-value |
|---|---|---|---|---|---|---|
| TNF-α | Protein | −1.84 | **0.0004** | 0.567 | F (2,33) = 21.60 | **1 × 10$^{-6}$** |
|  | Lipid | +1.18 | **0.0028** |  |  |  |
| IL-6 | Protein | −51.07 | **1.01 × 10$^{-5}$** | 0.462 | F (2,33) = 14.17 | **3.6 × 10$^{-5}$** |
|  | Lipid | −7.96 | 0.312 |  |  |  |
| IL-1β | Protein | −18.88 | **1.6 × 10$^{-6}$** | 0.617 | F (2,33) = 26.64 | **<1.3 × 10$^{-7}$** |
|  | Lipid | +4.45 | **0.092** |  |  |  |
| IL-10 | Protein | −0.11 | 0.115 | 0.082 | F (2,33) =1.47 | 0.245 |
|  | Lipid | −0.01 | 0.904 |  |  |  |

Significant associations (p < 0.05) are bold.

metabolism, and repair [48,49]. Plasma cytokine concentrations are influenced by multiple factors, including mouse strain, age, microbiota, housing conditions, diet, circadian rhythm and assay sensitivity [14,78,50]. Our baseline plasma cytokine results compare with those of a study that measured serum cytokines in Swiss mice fed high-fat or high-carbohydrate diets for 56 days using Luminex and reported TNF-α, IL-6, IL-1β, and IL-10 concentrations ranging from 3–8 pg/mL, 4–7 pg/mL, ~80 pg/mL, and 15–20 pg/mL, respectively [51]. Our results are also similar to those obtained in healthy adult C57BL/6 mice [52]. Baseline IL-1β concentrations were higher in our study, which may reflect differences in assay sensitivity, diet formulation, or feeding duration. These comparisons indicate that our pre-LPS cytokine levels fall within physiological ranges reported for healthy mice.

Following LPS challenge, all four cytokines significantly increased in plasma, confirming that LPS caused a systemic inflammatory response. Our findings are consistent with previous studies in mice where intraperitoneal injection of LPS at similar doses led to an increase in circulating cytokines [31,32].

LPS is a component of the outer membrane of gram-negative bacteria and is recognized by pattern recognition receptors such as Toll-like receptor 4 (TLR4) found on macrophages, dendritic cells, and other innate immune cells [11]. When LPS binds to TLR4, it activates adaptor proteins called myeloid differentiation primary response 88 (MyD88) and Toll/IL-1 receptor domain–containing adaptor inducing interferon-β (TRIF), which trigger two main signaling routes inside the cell [53].

The MyD88 pathway rapidly activates the IκB kinase (IKK) complex, leading to the movement of transcription factors nuclear factor kappa B (NF-κB) and activator protein 1 (AP-1) into the nucleus [54]. The TRIF pathway, on the other hand, activates interferon regulatory factor 3 (IRF3), leading to a more sustained response [55]. Together, these signaling events stimulate the production of pro-inflammatory cytokines such as TNF-α, IL-1β, and IL-6, as well as anti-inflammatory cytokines such as IL-10, and also activate mitogen-activated protein kinases (MAPKs) and protein kinase B (Akt), which regulate cytokine production and immune cell survival [12].

Interestingly, serum IL-6 levels were detectable only post LPS challenge, confirming its role as an acute-phase cytokine, which is secreted after TLR4-NF-κB signaling within hours of LPS exposure [56,57]. Our results thus suggest that IL-6 is minimally expressed under homeostatic conditions but upregulated during infection or inflammation.

The significantly higher concentrations of serum cytokines observed in all dietary groups following LPS challenge suggest an interaction between dietary macronutrients and innate immune signaling pathways. Macronutrients provide substrates for glycolysis and AMP-activated protein kinase (AMPK) activation pathways used by the energy-demanding inflammatory response to LPS [58]. Beyond the provision of energy, macronutrients and LPS activate overlapping intracellular signaling cascades such as NF-κB and the Akt–mTOR pathway, which are central to both metabolic regulation and inflammatory cytokine production [59,60]. Moreover, the lipid A moiety of LPS consists of mainly saturated fatty acids resembling dietary lipids [61] This structural similarity may explain why nutrient sensing and pathogen sensing systems such as TLR4 can be activated by both dietary components and microbial ligands, leading to similar proinflammatory outcomes [62].

## High-carbohydrate diets increase inflammatory cytokines

In the present study, mice fed high-carbohydrate diets exhibited elevated serum TNF-α, IL-1β, and IL-6 levels compared to those on high-protein diets, indicating that excessive carbohydrate intake may upregulate systemic inflammatory tone. Consistent with this observation, several studies have reported that chronic consumption of high-carbohydrate diets in both rodents and humans increases circulating and tissue proinflammatory cytokines [63–65]. These effects are partly attributed to activation of innate immune signaling pathways such as NF-κB and Mitogen-Activated Protein Kinase (MAPK), either directly or through oxidative stress and Myeloid Differentiation Primary Response Protein 88 (MYD88)-dependent mechanisms [66,67]. Our long-term feeding model supports these findings, as the pro-inflammatory effects of high-carbohydrate intake were evident even under unchallenged conditions, suggesting that excessive carbohydrate

consumption may sensitize the immune system and promote chronic low-grade inflammation. Additionally, the observed elevation of IL-10 may represent a compensatory anti-inflammatory response to increased TNF-α levels, or alternatively reflect insulin-mediated induction of IL-10 via the Akt–mTOR pathway in macrophages [60,64].

## High-fat diets increase inflammatory cytokines

Elevated serum concentrations of TNF-α, IL-1β, and IL-6 were also observed in mice fed high-lipid diets compared to those on high-protein diets, implying proinflammatory effects of excessive dietary fat. Notably, these elevations were evident both before and after LPS challenge, suggesting that lipid intake can modulate immune activity in both resting and activated states. While some studies report that chronic high-fat feeding can blunt LPS-induced inflammation via NF-κB downregulation, our results showed heightened cytokine responses, likely influenced by both the quantity and type of fat consumed [68]. Soybean oil, which was used as the lipid source, contains a high proportion of omega-6 polyunsaturated fatty acids (PUFAs), particularly linoleic acid, which has been linked to NF-κB activation and increased pro-inflammatory cytokine production [69,70]. Although PUFAs are often considered anti-inflammatory, omega-6-rich oils like soybean oil may skew the inflammatory balance, particularly when consumed in excess. Our findings are consistent with reports showing that dietary lipid overload promotes oxidative stress, mitochondrial dysfunction, and intestinal inflammation [61,67,71]. Thus, the elevated cytokine levels observed in high lipid groups likely reflect an interaction between lipid load, fatty acid profile, and immune signaling pathways.

## High-protein diets suppress inflammatory cytokines

The reduced cytokine levels in mice fed on high-protein diets support an anti-inflammatory role for dietary protein. This aligns with previous studies in animals where high-protein interventions reduced NF-κB activity and cytokine production [72–74]. Similarly, in humans, consumption of a high-protein diet causes reduced circulating IL-6 and TNF-α as well as blunted LPS-stimulated cytokine release [75,76]. In our model, however, purified casein and cysteine were used under isocaloric conditions, isolating protein effects without confounders such as energy restriction and whole food matrix components. This controlled approach revealed that protein alone, independent of energy deficit, can suppress systemic inflammation, likely via downregulation of NF-κB signaling [77,79]. Interestingly, while some research suggests low-protein diets can also reduce certain markers in obesity or increase the risk of allergic sensitization, our results indicate that in healthy mice, adequate to high protein intake directly dampens inflammatory reactivity, even upon immune challenge [76,79]. These results show that not just macronutrient balance, but protein quantity and quality shape the immune system.

Finally, the lower cytokine responses observed in the high-protein diet groups may represent a double-edged effect. In the context of acute inflammation, such as an LPS challenge, a reduced cytokine response could indicate impaired immune activation and a diminished ability to eliminate invading agents, consistent with reports that associate high-protein intake with adverse metabolic and immune outcomes [80,81]. However, in chronic inflammatory states, a moderated cytokine response might be advantageous by limiting excessive inflammation and reducing tissue damage [14]. This duality highlights that the physiological consequences of attenuated cytokine activity depend on the nature and duration of the inflammatory challenge. Taken together, these findings suggest that dietary protein influences immune homeostasis in a context-dependent manner, with potential to both benefit and compromise health depending on the underlying inflammatory context.

## Protein and lipid intake predict cytokine changes post-LPS

Multiple regression analyses showed a strong association of dietary protein and fold change of serum TNF-α, IL-6, and IL-1β levels following LPS challenge, indicating a possible anti-inflammatory effect. This agrees with evidence that dietary

protein may mitigate excessive inflammatory responses [74,78,38]. In contrast, increased dietary lipid significantly elevated TNF-α responses but did not significantly influence IL-6 or IL-1β, suggesting a selective pro-inflammatory role of lipids, potentially through mechanisms such as enhanced TLR4 signaling [82,83]. Our results imply that high-carbohydrate, low-protein diets are associated with increased pro-inflammatory cytokine responses, consistent with reports that excessive carbohydrate intake can potentiate inflammation [64,65]. Interestingly, neither protein nor lipid intake significantly predicted IL-10 responses, indicating that the anti-inflammatory arm of the cytokine response might be influenced by factors beyond macronutrient balance.

In conclusion, by using purified isocaloric diets in healthy mice, this study has shown that chronic consumption of high-carbohydrate and high-lipid diets promotes a pro-inflammatory environment, while high-protein diets exert anti-inflammatory effects. Hence systemic inflammation may be modulated by reducing dietary carbohydrate and lipid while maintaining adequate protein intake. Importantly, the rapid activation of key inflammatory cytokines following LPS challenge likely reflects an enhanced innate immune responsiveness influenced by dietary composition. While such prompt activation is essential for effective host defense, persistent or exaggerated responses could also predispose to chronic inflammation, pointing to the dual nature of cytokine activation in immune regulation.

## 5 Limitations and future directions

While the use of purified isocaloric diets allowed precise control of macronutrient composition in our study, it does not fully reflect the complexity of whole food diets, which contain many nutrients and bioactive compounds that may also influence immune responses.

These results reflect responses in healthy male and female Swiss albino mice and may not reflect responses in aged or disease models such as obesity or diabetes. Although both male and female mice were included to enhance generalizability and the group size was n = 6, the sample size per sex of n = 3 reduced the power to detect sex-specific effects on cytokine responses. This study did not assess gut microbiota composition, which is known to be influenced by dietary macronutrients and can modulate systemic inflammation. Similarly, cytokine levels were measured at a single timepoint, limiting our understanding of dynamic immune responses. Our study, lacks a control group like many animal studies. This is due to the fact that the AIN-93M diet, which is a standard reference diet for rodents, is itself a high-carbohydrate diet and therefore could not serve as a balanced comparator. Finally, although all mice were provided with equal rations that were nearly completely consumed, individual food intake was not measured. Therefore, small differences in individual consumption cannot be entirely excluded.

Future studies should include an evenly distributed macronutrient control diet to provide a neutral baseline for evaluating the effects of extreme nutrient ratios. Additional work could also explore how specific amino acid profiles or fatty acid subtypes influence inflammatory pathways over time, incorporate microbiota analysis, and assess functional immune outcomes such as infection resistance or vaccine response. Studies with larger sex-stratified groups and disease models would also improve translational relevance.

## Supporting information

**S1 File. Data sets used in the study.**
(PDF)

## Acknowledgments

The authors would like to acknowledge and appreciate the support of Mr. Charles Kamusiime for handling the animals at the animal house. We also acknowledge the assistance of Mr. Ronald Kizza and Agnes Turyamubona in the laboratory phase of the study.

## AI use declaration

ChatGPT (Open AI, 2025) was used to improve the clarity and language of the manuscript. The authors declare that the hypotheses, interpretations, results and conclusions presented are entirely their own. No data generation, analysis or interpretation was performed by AI.

## Author contributions

**Conceptualization:** Hellen W. Kinyi, Charles Kato Drago, Lucy Ochola, Gertrude N. Kiwanuka.

**Data curation:** Hellen W. Kinyi, Gertrude N. Kiwanuka.

**Formal analysis:** Hellen W. Kinyi, Charles Kato Drago, Gertrude N. Kiwanuka.

**Investigation:** Hellen W. Kinyi.

**Methodology:** Hellen W. Kinyi, Charles Kato Drago, Lucy Ochola, Gertrude N. Kiwanuka.

**Supervision:** Charles Kato Drago, Lucy Ochola, Gertrude N. Kiwanuka.

**Validation:** Hellen W. Kinyi, Gertrude N. Kiwanuka.

**Visualization:** Hellen W. Kinyi.

**Writing – original draft:** Hellen W. Kinyi.

**Writing – review & editing:** Hellen W. Kinyi, Charles Kato Drago, Lucy Ochola, Gertrude N. Kiwanuka.

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
