## [Decision Letter · Decision Letter 0]

24 Sep 2025

Dear Dr. Kinyi,

Thank you for submitting your manuscript to PLOS ONE. After careful consideration, we feel that it has merit but does not fully meet PLOS ONE’s publication criteria as it currently stands. Therefore, we invite you to submit a revised version of the manuscript that addresses the points raised during the review process.

We look forward to receiving your revised manuscript.

Kind regards,

Rami Salim Najjar, Ph.D.

Academic Editor

PLOS ONE

Journal Requirements:

2. We note that your Data Availability Statement is currently as follows: All relevant data are within the manuscript and in Supporting Information files.

3. Please ensure that you refer to Figure 4 in your text as, if accepted, production will need this reference to link the reader to the figure.

Reviewers' comments:

Reviewer's Responses to Questions

**Comments to the Author**

1. Is the manuscript technically sound, and do the data support the conclusions?

Reviewer #1: Partly

Reviewer #2: Yes

2. Has the statistical analysis been performed appropriately and rigorously?

Reviewer #1: Yes

Reviewer #2: Yes

3. Have the authors made all data underlying the findings in their manuscript fully available?

Reviewer #1: Yes

Reviewer #2: Yes

4. Is the manuscript presented in an intelligible fashion and written in standard English?

Reviewer #1: Yes

Reviewer #2: Yes

Reviewer #1: Title: High-protein diets reduce plasma pro-inflammatory cytokines following lipopolysaccharide challenge in Swiss Albino mice

Overall, the authors present a well-written manuscript addressing the need to elucidate the impact of macronutrient composition on the innate immune response. This reviewer has a few comments and questions:

• The authors discuss the small sample size (n=3 per sex) in their limitations; however, in 2.2.2 and in Figure 1, the sample size is noted to be six male and six female. Furthermore, the sample size is not indicated in the captions of the results figures. Was there a loss in the samples?

• A high-carbohydrate diet for 15 weeks can induce diabetes in mice such as C57s, was blood glucose measured in these animals? Did any animals exhibit signs of diabetes?

• Can the authors provide rationale for not including a control group in which the macronutrient composition follows the standard recommendations? Or add it to the limitations?

• Did the authors measure food intake? If so, it should be included alongside the changes in weight results.

• Can the authors discuss any statistics on human consumption patterns that may align with the macronutrient compositions discussed here? For example, HLLC closely resembles a more extreme ketogenic diet.

• Other comments:

o On line 254: Authors may be referring to Figure 2, not Figure 1.

o Lines 393 and 409: Please consider following the same phrasing for both or combining them into one section.

o Periods are often missing throughout the manuscript after citations.

Reviewer #2: The manuscript of Kinyi et al assessed macronutrient induced differences in cytokine induction following a challenge with LPS. This is a highly interesting and important topic of research as food plays a tremendous role in the regulation of health, including the immune system. The manuscript and methods are well described and results technically sound.

My prime concern however, is about the interpretation of research findings. In particular, whether the rapid activation of four cytokines can be considered as a good or a bad sign.

First, the authors describe subtle but significant alterations of plasma cytokine levels in unchallenged mice fed various well-balanced diets. As all animals are still of a very young age, it would be important to indicate which concentrations can still be considered within the normal range and which indeed exceed a certain threshold and can truly be considered elevated e.g., by measuring cytokine levels of control mice on the standard AIN93M-food or using reference values from literature.

Second, they expose animals on diverse isocaloric diets to LPS and monitor the subsequent activation of cytokines 3 hours later. Here, I would rather suggest these increased levels to be an indication of how well the different animals can still cope with and respond to stress. This should be illustrated as additional figure according to as described by the authors in their methods section line 240-241: “Change in serum cytokine levels (� cytokine = Post LPS challenge level – Pro LPS challenge level) was calculated for each mouse.”

Third, both high protein diets show less of an increase compared to the other diets, suggesting a reduced responsiveness. This scenario, in contrast to what is currently indicated in the discussion, would be fully in line with adverse health conditions driven by high protein diets and improved health by low protein diets (for example see Solon-Biet SM et al 2015 Proc Natl Acad Sci U S A DOI: 10.1073/pnas.1422041112, Kitada M et al 2019 EBioMedicine DOI: 10.1016/j.ebiom.2019.04.005, and van Galen I et al 2025 NPJ Metab Health Dis DOI: 10.1038/s44324-025-00064-3) and elaboration thereof within the discussion in light of the present cytokine date would further strengthen this manuscript.

Fourth, the authors might want to reconsider changing the overall conclusions such as on line 37 “high dietary protein intake reduced pro-inflammatory responses” more into high dietary restriction dampened the responsiveness towards lipopolysaccharide challenge, as indicated by less increased pro-inflammatory cytokine levels. Also overstatements such as on line 357-358 “control mice fed on high-protein diets exhibited lower cytokine levels” should be avoided as this was not significantly lower and likely within the normal range.

Fifth, many cytokines are under the control of circadian rhythm such as TNFa, IL6 and IL18. Was there any difference or correlation of cytokine levels with collection time or deviations between groups?

Additional minor concerns are:

- The authors should systematically correct textual and grammar issues e.g., dots are missing at the end of some sentences, a mixture of US and UK Engligh is used, reference to figures is frequently wrong and the reference list shows many inconsistencies.

- In Table 1 also the basal CHO, Protein and Lipid levels of AIN93M food should be indicated along with indications on how much these levels are changed in the various diets. Also, I assume the percentage composition equals a macronutrient ratio as % Kcal.

- Line 133-134 indicates that only animals that fulfilled the inclusion criteria were randomized across groups. Which inclusion criteria were used?

- Line 252-254 reports on inconsistent highest and lowest diet groups compared to the data in Figure 2.

- As male and female mice can have large differences in body weight and show some metabolic differences, the question arises if the diet induced changes in body weight gain over 15 weeks are consistent in both sexes. Showing this as e.g. supplemental separate graph for males and females would be of support.

- All graphs show error bars without indication of what type of error was used. Is this SD, SEM, or 95% CI?

- The manuscript would overall gain from some further mechanistic analysis by characterization of molecular changes of the pathways indicated following LPS challenge on the various diets.

- Lastly, the authors should add any conflict of interest statement and indicate if and how AI was used.

**Do you want your identity to be public for this peer review?** For information about this choice, including consent withdrawal, please see our Privacy Policy

Reviewer #1: No

Reviewer #2: No

---

## [Author Response · Author response to Decision Letter 1]

5 Nov 2025

Editors Comments

1. Ensure that your manuscript meets PLOS ONE's style requirements, including those for file naming.

-Author affiliations corrections made: Removed the PO BOX addresses, and added abbreviations (HWK) for the corresponding author.

-Title and short title have been Italicized and bolded and written in in sentence case

2. Data availability statement. We note that your Data Availability Statement is currently as follows: All relevant data are within the manuscript and in Supporting Information files. Please confirm at this time whether or not your submission contains all raw data required to replicate the results of your study.

We confirm that all data required to replicate the findings of this study are included within the manuscript and its Supporting Information files.

3. Figure 4. Seems this figure has not been cited in the text. Please ensure that you refer to Figure 4 in your text as, if accepted, production will need this reference to link the reader to the figure.

Thanks for this correction. Figure 4 has been cited in section 3.2 Line 306. The change has been highlighted.

4. Supporting information. Please include captions for your Supporting Information files at the end of your manuscript, and update any in-text citations to match accordingly.

Captions for supporting information have been added at the end of the manuscript as suggested (Page 27, Lines 835-843). The supporting information has been highlighted.

Reviewer 1 comments

We thank Reviewer 1 for their time and thoughtful evaluation of our manuscript. We greatly appreciate their positive comments recognizing the clarity of the manuscript and the importance of investigating how macronutrient composition influences innate immune responses. We have carefully considered all comments and have addressed them point by point below.

1. The authors discuss the small sample size (n=3 per sex) in their limitations; however, in 2.2.2 and in Figure 1, the sample size is noted to be six male and six female. Furthermore, the sample size is not indicated in the captions of the results figures. Was there a loss in the samples?

We thank reviewer 1 for this observation and the opportunity to clarify. No animals were lost during the study.

Each experimental group consisted of six animals (n=6), comprising three males and three females. Thus, for the purposes of statistical analyses, the group sample size was n=6. However, as noted in the limitations, when stratified by sex, the sample size was effectively n=3 per sex, which reduced the power to detect sex-specific effects. To avoid confusion, we have now clarified this in the Methods (Section 2.2.2, lines 130-131) and Limitation (Section 5.0, lines 533-536) sections and added the group sample size to all figure captions. These have been highlighted in yellow.

2. A high-carbohydrate diet for 15 weeks can induce diabetes in mice such as C57s, was blood glucose measured in these animals? Did any animals exhibit signs of diabetes?

We thank reviewer 1 for this observation. In the present study, our primary focus was the immune response to LPS challenge, and therefore blood glucose was not measured. However, in our previous work using the same dietary formulations in Swiss albino mice, we assessed blood glucose and found only mild hyperglycaemia under random feeding conditions in the HCLP group and fasting hypoglycaemia in the HLLP group, both within physiological ranges for this strain ((Kinyi et al. 2025, Food Science & Nutrition, 13(9): e70957). While C57BL/6J mice are well known for their high susceptibility to diet-induced diabetes, Swiss albino mice did not show such responses. Additionally, mice in the wild typically consume grain- and plant-based diets that are naturally rich in carbohydrates. The AIN-93M formulation reflects this nutritional pattern by providing carbohydrates as the main energy source. Thus, the high-carbohydrate content of some of our experimental diets is consistent with the species’ natural dietary tendencies.

3. Can the authors provide rationale for not including a control group in which the macronutrient composition follows the standard recommendations? Or add it to the limitations?

We thank reviewer 1 for this valuable comment. Our work was modelled around the AIN-93M diet, which is the standard reference formulation recommended by the American Institute of Nutrition for rodent studies. However, we acknowledge that AIN-93M is itself a high-carbohydrate formulation (76% carbohydrate, 15% protein, 9% lipid), and therefore its macronutrient profile is somehow similar to our HCLL experimental diet (75% CHO, 20% protein, 5% lipid). Although the AIN-93M diet is used as the standard reference, its high-carbohydrate profile limited its suitability as a neutral control, given that our study also included separate high-carbohydrate, high-lipid, and high-protein diets. We have added this point in the Limitations section of the manuscript Section 5.0 (Lines 539–541).

4. Did the authors measure food intake? If so, it should be included alongside the changes in weight results. We appreciate this comment regarding food intake.

In this study, a fixed ratio of 5 g/mouse/day of diet was provided per cage daily, which is within the normal daily intake range for Swiss albino mice. Few leftovers were observed upon daily inspection. Therefore, while individual food intake was not quantified, all groups had equal access to food and water (provided ad libitum). This has been clarified in the Methods section (Lines 156-160). We have also acknowledged in the Limitations section that the absence of individual food intake measurements could introduce minor variability in interpreting diet-related effects on body weight and immune responses (Lines 541-544).

5. Can the authors discuss any statistics on human consumption patterns that may align with the macronutrient compositions discussed here? For example, HLLC closely resembles a more extreme ketogenic diet.

We thank reviewer 1 for this valuable suggestion. To strengthen the translational relevance of our findings, we have now included a discussion of how the experimental diets parallel certain human dietary patterns (Section 4.0, Lines 381–388).

6. On line 254: Authors may be referring to Figure 2, not Figure 1.

Thanks for this correction. We have indicated the correct figure.

7. Lines 393 and 409: Please consider following the same phrasing for both or combining them into one section. We thank reviewer 1 for this helpful observation. We understand how the opening statement of the high-carbohydrate section, which referred to both high-carbohydrate and high-lipid diets, may have created the impression that these sections could be merged. To address this, we have revised the introductory sentences to make the focus of each section clearer and specific to the respective diet type. In addition, we have modified the section titles so that they follow a parallel structure, thereby improving clarity and consistency. (Lines 450–452 and 464–467).

8. Periods are often missing throughout the manuscript after citations.

We sincerely apologize for this anomaly. We have added periods at the end of every sentence.

Reviewer 2

We sincerely thank Reviewer 2 for their time, thoughtful evaluation, and encouraging remarks on our manuscript. We greatly appreciate the recognition of the importance of our work on macronutrient-induced differences in cytokine responses to LPS challenge, as well as the positive feedback regarding the clarity of our methods and the soundness of our results. We have carefully considered all comments and suggestions provided and have addressed each point in detail below.

1. My prime concern however, is about the interpretation of research findings. In particular, whether the rapid activation of four cytokines can be considered as a good or a bad sign.

We thank the reviewer for this insightful comment. We agree that interpreting the biological significance of rapid cytokine activation is important for understanding the implications of our findings. In response, we have added text to the Conclusion section (Lines 523–527) to clarify that early cytokine activation likely reflects enhanced innate immune responsiveness, which is beneficial for host defence, but that excessive or prolonged activation could also contribute to chronic inflammation. We hope this addition provides a more balanced interpretation of the observed response patterns.

2. First, the authors describe subtle but significant alterations of plasma cytokine levels in unchallenged mice fed various well-balanced diets. As all animals are still of a very young age, it would be important to indicate which concentrations can still be considered within the normal range and which indeed exceed a certain threshold and can truly be considered elevated e.g., by measuring cytokine levels of control mice on the standard AIN93M-food or using reference values from literature.

We thank Reviewer 2 for this comment. We have now addressed this by comparing our baseline cytokine concentrations to values reported in two previous studies conducted in healthy, unchallenged mice. First, de Almeida-Souza et al. (2018) measured serum cytokines in Swiss mice fed high-fat or high-carbohydrate diets for 56 days using Luminex and second, Dogan et al. (2017) reported baseline TNF-α, IL-6, IL-1β, and IL-10 levels in healthy adult C57BL/6 mice. Our baseline TNF-α, IL-6, and IL-10 concentrations fall within the ranges reported in these studies, indicating that the cytokine levels observed in our study reflect physiological baseline values rather than pathological elevations. This comparison has been added to the Discussion section (Lines 407–415).

3. Second, they expose animals on diverse isocaloric diets to LPS and monitor the subsequent activation of cytokines 3 hours later. Here, I would rather suggest these increased levels to be an indication of how well the different animals can still cope with and respond to stress. This should be illustrated as additional figure according to as described by the authors in their methods section line 240-241: “Change in serum cytokine levels (D cytokine = Post LPS challenge level – Pro LPS challenge level) was calculated for each mouse.”

We thank the reviewer 2 for this helpful suggestion. We agree that illustrating the change in cytokine levels following LPS challenge provides a clearer picture of the animals’ capacity to respond to stress. In response, we have included an additional figure (now Figure 5) illustrating the mean (± SEM) change in plasma cytokine concentrations 3 hours after LPS. that visualizes change in cytokine levels across diets. We have also added a new Results subsection (Section 3.3) describing these changes in detail.

4. Third, both high protein diets show less of an increase compared to the other diets, suggesting a reduced responsiveness. This scenario, in contrast to what is currently indicated in the discussion, would be fully in line with adverse health conditions driven by high protein diets and improved health by low protein diets (for example see Solon-Biet SM et al 2015 Proc Natl Acad Sci U S A DOI: 10.1073/pnas.1422041112, Kitada M et al 2019 EBioMedicine DOI: 10.1016/j.ebiom.2019.04.005, and van Galen I et al 2025 NPJ Metab Health Dis DOI: 10.1038/s44324-025-00064-3) and elaboration thereof within the discussion in light of the present cytokine date would further strengthen this manuscript.

We thank the reviewer for this comment and for the helpful references provided. We fully agree that the reduced cytokine responsiveness observed in the high-protein diet groups could reflect an adverse outcome, consistent with reports linking high-protein intake to metabolic stress and impaired immune regulation (Solon-Biet et al., 2015; Kitada et al., 2019; van Galen et al., 2025). In line with the reviewer’s observation, we have expanded the Discussion (Lines 496–506) to include this interpretation and its relevance to our findings. We also note that, depending on the physiological context, blunted cytokine responses may not always be detrimental. In chronic inflammatory states, for example, moderate suppression of cytokine production has been associated with reduced tissue injury and improved metabolic outcomes (e.g., Mishra 2024). We have incorporated this point to emphasize that the observed effect of high-protein diets may reflect a complex, context-dependent modulation of immune responsiveness.

5. Fourth, the authors might want to reconsider changing the overall conclusions such as on line 37 “high dietary protein intake reduced pro-inflammatory responses” more into high dietary restriction dampened the responsiveness towards lipopolysaccharide challenge, as indicated by less increased pro-inflammatory cytokine levels. Also, overstatements such as on line 357-358 “control mice fed on high-protein diets exhibited lower cytokine levels” should be avoided as this was not significantly lower and likely within the normal range. We thank Reviewer 2 for this comment.

We have revised the statement as suggested (Lines 39-42). The new sentence clarifies the comparison between high- and low-protein diets. We have also included language to indicate this reflects a relative dampening of cytokine responsiveness, aligning with the reviewer’s concern (Lines 393-395).

6. Fifth, many cytokines are under the control of circadian rhythm such as TNFa, IL6 and IL18. Was there any difference or correlation of cytokine levels with collection time or deviations between groups?

All mice were euthanised within the same time window between 9:00 a.m. and 12:00 p.m. to minimize potential effects of circadian variation. Consequently, we did not monitor or analyse circadian fluctuations in cytokine levels. This standardized timing ensured that any differences observed among groups were attributable to dietary or treatment effects rather than time-of-day–related variations.

7. The authors should systematically correct textual and grammar issues e.g., dots are missing at the end of some sentences, a mixture of US and UK English is used, reference to figures is frequently wrong and the reference list shows many inconsistencies.

We thank the reviewer for this valuable feedback. The entire manuscript has been carefully proofread to correct grammatical and punctuation errors, ensure consistent use of UK English throughout, and verify all figure and reference citations. The reference list has also been reviewed and reformatted to ensure consistency with journal requirements.

8. In Table 1 also the basal CHO, Protein and Lipid levels of AIN93M food should be indicated along with indications on how much these levels are changed in the various diets. Also, I assume the percentage composition equals a macronutrient ratio as % Kcal.

We thank the reviewer for this helpful suggestion. The experimental diets in this study were formulated as modified versions of the AIN-93M reference diet; however, the standard AIN-93M diet itself was not included as an experimental group. To avoid giving the impression that it was part of the tested diets, we have added the percentage composition of macronutrients in the AIN-93M diet in the text under the Materials and Methods section, rather than in Table 1 (Lines 96-98). We also confirm that the macronutrient percentages presented in Table 1 represent the proportion of total dietary energy contributed by each macronutrient (% kcal). Each diet was designed to be isocaloric at 3.8 kcal/g. The table caption has been revised accordingly for clarity.

9. Line 133-134 indicates that only animals that fulfilled the inclusion criteria were randomized across groups. Which inclusion criteria were used?

We thank the reviewer for this comment. The inclusion criteria have now been clearly stated in the revised text. Only healthy Swiss albino mice aged 6–8 weeks and weighing 18–23 g were included in the study prior to randomization (Lines 133-137).

10. Line 252-254 reports on inconsistent highest and lowest diet groups compared to the data in Figure 2. We thank the reviewer for noting this inconsistency. Upon review, we discovered that the values for the HPLC and HCLP groups had been inadvertently interchanged durin

---

## [Decision Letter · Decision Letter 1]

25 Nov 2025

High-protein diets reduce plasma pro-inflammatory cytokines following lipopolysaccharide challenge in Swiss Albino mice

PONE-D-25-44328R1

Dear Dr. Kinyi,

We’re pleased to inform you that your manuscript has been judged scientifically suitable for publication and will be formally accepted for publication once it meets all outstanding technical requirements.

Kind regards,

Rami Salim Najjar, Ph.D.

Academic Editor

PLOS ONE

Additional Editor Comments (optional):

Reviewers' comments:

Reviewer's Responses to Questions

**Comments to the Author**

Reviewer #1: All comments have been addressed

2. Is the manuscript technically sound, and do the data support the conclusions?

Reviewer #1: Yes

3. Has the statistical analysis been performed appropriately and rigorously?

Reviewer #1: Yes

4. Have the authors made all data underlying the findings in their manuscript fully available?

Reviewer #1: Yes

5. Is the manuscript presented in an intelligible fashion and written in standard English?

Reviewer #1: Yes

Reviewer #1: Thank you to the authors for thoroughly addressing my previous comments and for the thoughtful revisions to their manuscript.

**Do you want your identity to be public for this peer review?** For information about this choice, including consent withdrawal, please see our Privacy Policy

Reviewer #1: No

---

## [Editor Report · Acceptance letter]

PONE-D-25-44328R1

PLOS One

Dear Dr. Kinyi,

I'm pleased to inform you that your manuscript has been deemed suitable for publication in PLOS One. Congratulations! Your manuscript is now being handed over to our production team.

Kind regards,

on behalf of

Dr. Rami Salim Najjar

Academic Editor

PLOS One